# Projected impact of a reduction in sugar-sweetened beverage consumption on diabetes and cardiovascular disease in Argentina: A modeling study

**M. Victoria Salgado** [1] *, **Joanne Penko**[2,3], **Alicia Fernandez**[4], **Jonatan Konfino**[1], **Pamela G. Coxson** [2,3], **Kirsten Bibbins-Domingo**[2,3,4], **Raul Mejia**[1]

**1** Centro de Estudios de Estado y Sociedad (CEDES), Ciudad de Buenos Aires, Argentina, **2** Center for Vulnerable Populations, University of California San Francisco, San Francisco, California, United States of America, **3** Department of Epidemiology and Biostatistics, University of California San Francisco, San Francisco, California, United States of America, **4** Department of Medicine, University of California San Francisco, San Francisco, California, United States of America

* victoria.salgado@cedes.org

**Data Availability Statement:** Data for this study come from sources detailed in the paper. Data on CVD risk factors come from the CESCAS I study

## Abstract

### Background

Sugar-sweetened beverage (SSB) consumption is associated with obesity, diabetes, and hypertension. Argentina is one of the major consumers of SSBs per capita worldwide. Determining the impact of SSB reduction on health will inform policy debates.

### Methods and findings

We used the Cardiovascular Disease Policy Model-Argentina (CVD Policy Model-Argentina), a local adaptation of a well-established computer simulation model that projects cardiovascular and mortality events for the population 35–94 years old, to estimate the impact of reducing SSB consumption on diabetes incidence, cardiovascular events, and mortality in Argentina during the period 2015–2024, using local demographic and consumption data. Given uncertainty regarding the exact amount of SSBs consumed by different age groups, we modeled 2 estimates of baseline consumption (low and high) under 2 different scenarios: a 10% and a 20% decrease in SSB consumption. We also included a range of caloric compensation in the model (0%, 39%, and 100%). We used Monte Carlo simulations to generate 95% uncertainty intervals (UIs) around our primary outcome measures for each intervention scenario. Over the 2015–2024 period, a 10% reduction in SSBs with a caloric compensation of 39% is projected to reduce incident diabetes cases by 13,300 (95% UI 10,800–15,600 [low SSB consumption estimate]) to 27,700 cases (95% UI 22,400–32,400 [high SSB consumption estimate]), i.e., 1.7% and 3.6% fewer cases, respectively, compared to a scenario of no change in SSB consumption. It would also reduce myocardial infarctions by 2,500 (95% UI 2,200–2,800) to 5,100 (95% UI 4,500–5,700) events and all-cause deaths by 2,700 (95% UI 2,200–3,200) to 5,600 (95% UI 4,600–6,600) for "low" and "high" estimates of SSB intake, respectively. A 20% reduction in SSB consumption with 39% caloric compensation is

(https://estudiocescas.iecs.org.ar/), the 2013 National Risk Factor Survey (http://www.msal.gob. ar/ent/index.php/vigilancia/publicaciones/ encuestas-poblacionales), and the PrEViSTA study (https://www.ncbi.nlm.nih.gov/pubmed/ 24024917). Framingham Heart Study data are available following approval of research applications submitted through the National Heart, Lung, and Blood Institute's Biologic Specimen and Data Repository Information Coordinating Center (available at http://biolincc.nhlbi.nih.gov/studies/ framcohort/?q=framingham for the Framingham Cohort and http://biolincc.nhlbi.nih.gov/studies/ framoffspring/?q=framingham for the Offspring study). Data on sugar sweetened beverage consumption come from CESCAS I study (https:// estudiocescas.iecs.org.ar/), the 2005 National Nutrition and Health Survey (http://www.msal.gob. ar/images/stories/bes/graficos/0000000257cnt- a08-ennys-documento-de-resultados-2007.pdf), and Euromonitor (https://www.euromonitor.com/ argentina). Vital statistics and census data are publicly available from government sources described in the paper.

**Funding:** RM Grant 108168-001 International Development Research Centre (IDRC), Canada https://www.idrc.ca/ The funders had no role in study design, data collection and analysis, decision to publish, or preparation of the manuscript.

**Competing interests:** The authors have declared that no competing interests exist.

**Abbreviations:** BMI, body mass index; CHD, coronary heart disease; CVD Policy Model- Argentina, Cardiovascular Disease Policy Model- Argentina; CVD, cardiovascular disease; ENNyS, Encuesta Nacional de Nutrición y Salud; PrEViSTA, Program for the Epidemiological Evaluation of Stroke in Tandil; SBP, systolic blood pressure; SSB, sugar-sweetened beverage; STROBE, Strengthening the Reporting of Observational Studies in Epidemiology; UI, uncertainty interval.

projected to result in 26,200 (95% UI 21,200–30,600) to 53,800 (95% UI 43,900–62,700) fewer cases of diabetes, 4,800 (95% UI 4,200–5,300) to 10,000 (95% UI 8,800–11,200) fewer myocardial infarctions, and 5,200 (95% UI 4,300–6,200) to 11,000 (95% UI 9,100– 13,100) fewer deaths. The largest reductions in diabetes and cardiovascular events were observed in the youngest age group modeled (35–44 years) for both men and women; additionally, more events could be avoided in men compared to women in all age groups. The main limitations of our study are the limited availability of SSB consumption data in Argentina and the fact that we were only able to model the possible benefits of the interventions for the population older than 34 years.

## Conclusions

Our study finds that, even under conservative assumptions, a relatively small reduction in SSB consumption could lead to a substantial decrease in diabetes incidence, cardiovascular events, and mortality in Argentina.

## Author summary

### Why was this study done?

- Sugar-sweetened beverages (SSBs) are associated with obesity, hypertension, and diabetes.

- Argentina is one of the largest consumers of SSBs, particularly sodas, in the world.

- When considering measures aimed at reducing SSB consumption, such as a soda tax, policy makers need evidence-based estimates of potential health benefits.

### What did the researchers do and find?

- We first developed and updated an Argentinian version of the Cardiovascular Disease Policy Model (CVD Policy Model), a well-established computer simulation model already used in the United States and Mexico to estimate cardiovascular health outcomes.

- We used the CVD Policy Model-Argentina to determine the potential impact of a reduction in soda consumption on diabetes, cardiovascular diseases (CVDs), and mortality among adults 35–94 years of age over a 10-year period (2015–2024).

- A 10% reduction in soda consumption is projected to avert between 13,300 to 27,700 diabetes cases, 2,500 to 5,100 myocardial infarctions, and 2,700 to 5,600 all-cause deaths over a 10-year period.

- The largest reductions in diabetes and cardiovascular events were observed in the youngest age group modeled (35–44 years) for both men and women; additionally, more events could be avoided in men compared to women in all age groups.

**What do these findings mean?**

- A relatively small reduction in SSB consumption could lead to a substantial decrease in diabetes incidence, cardiovascular events, and mortality in Argentina.

- These results support the implementation of policies to reduce SSB consumption, such as a soda tax. Use of taxation as a health policy tool would have the additional advantage of providing a new source of public funds to support healthy lifestyles.

## Introduction

As one of the main sources of added sugar in Western diets, sugar-sweetened beverage (SSB) consumption is suggestive of poor dietary quality [1] and is associated with obesity, type 2 diabetes mellitus (from now on, referred to only as "diabetes"), and hypertension [2–8]. While sales of sweet soft drinks in North America leveled off after 2012, in Latin America they doubled in the period 2000–2013 [9]. Increased SSB consumption during this time period was strongly correlated with growing rates of overweight and obesity in the region [9].

Argentina is one of the world's highest consumers of SSBs, with consumption estimates of 120 to 130 liters of SSB per capita per year [10,11]. Between 2005 and 2013, the combined overweight and obesity prevalence in Argentina rose from 49% to 58%; 21% of the population presented obesity in 2013 [12]. This increase in obesity has contributed to rising rates of diabetes —diagnosed in 9.8% of Argentines in 2013 [12]—and to Argentina's very high rates of cardiovascular disease (CVD) [9]. Given that relatively small increases in weight heighten the risk of diabetes and CVD [13–16], SSBs may contribute to disease development even among individuals without obesity.

Due to the increase in obesity prevalence and its related illnesses, reducing SSB consumption—which in Argentina consists overwhelmingly of sugary sodas [10]—is a public health priority. The World Health Organization has suggested SSB taxation as a fiscal policy intervention for the prevention of noncommunicable diseases [17]. The success of Mexico's SSB tax, which led to an 11% price increase followed by a 7.3% reduction in SSB sales within the first 2 years, has placed SSB taxes on the menu of policy options for all Latin American countries [18].

In order to provide local policy makers estimates of the projected impact of SSB taxation on the health of the Argentine population, we used a well-established computer simulation model, the Cardiovascular Disease Policy Model (CVD Policy Model), adapted for the Argentine population, to simulate the impact of reduced SSB consumption on national diabetes incidence, cardiovascular events, and mortality.

## Methods

### CVD Policy Model-Argentina

The CVD Policy Model is a computer simulation, state transition (Markov) model that estimates the prevalence and incidence of CVD by using demographic, epidemiological, vital statistic, and hospital data measured in the population 35 years old and older. The model separates the population into those without and with CVD. Those without CVD are stratified into cells defined by sex, age decile, and levels of the following cardiovascular risk factors: systolic

blood pressure (SBP; <130; 130–139.9; ≥140 mmHg), low-density lipoprotein cholesterol (<100; 100–129.9; ≥130 mg/dl), high-density lipoprotein cholesterol (<40; 40–59.9; ≥60 mg/dl), smoking status (no exposure, second-hand smoke exposure, active smoking), type 2 diabetes status (yes versus no), and body mass index (BMI) (<25; 25–29.9; ≥30 kg/m$^2$). In annual cycles, those without preexisting CVD have probabilities of experiencing incident coronary heart disease (CHD), incident stroke, or death from non-cardiovascular causes, with transition rates dependent on age, sex, and risk factor values. The population with prior CVD has annual rates of recurrent cardiovascular events or death from cardiovascular or non-cardiovascular causes, with transition rates dependent on age, sex, and prior CVD status. Each annual cycle, new 35-year-olds enter the simulated population, measured from census projections [19,20], and those who die or reach 95 years of age exit the simulated population.

The first version of the CVD Policy Model-Argentina was developed in 2009 [21]. Since then, new sources of information have become available and have replaced original inputs, including the 2010 National Census [22–24], the 2013 National Risk Factor Survey [12], the Study for the Detection and Follow-up of Cardiovascular Disease Risk Factors in the Southern Cone of Latin America (CESCAS I study) [25], and the Program for the Epidemiological Evaluation of Stroke in Tandil (PrEViSTA) [26]. The updated version of the model was calibrated with an accuracy of 99.5% when comparing CVD events and deaths predicted by this model and those observed in national data for 2010 [27]. A more detailed explanation of model development and update can be found in the S1 Appendix, as well as in a previous publication [27].

The study's prospective protocol can be found in the S1 Protocol. This study is reported as per the Strengthening the Reporting of Observational Studies in Epidemiology (STROBE) guideline (S1 STROBE Checklist).

## Estimating daily SSB consumption in Argentina

To model the impact of a decrease in SSB consumption on cardiovascular health, we first estimated per capita daily consumption of SSBs in Argentina by sex and age group. We focused specifically on sugar-sweetened soda consumption (hereafter labeled "SSB") and estimated mean daily 12-ounce servings of these beverages. This serving size is equivalent to a can of soda in Argentina. Due to a lack of a single source of nationally representative data on SSB consumption for each age and sex stratum modeled, we generated 2 sets of estimates (a "low" estimate and a "high" estimate) using the best available data and modeled both scenarios to represent the range of likely SSB intake in the population.

We obtained our "low" estimates from the CESCAS I study, an ongoing observational prospective cohort designed to study CVD prevalence and risk factors in Southern Latin America. In 2010–2011, CESCAS I measured self-reported average daily SSB consumption among 3,300 adults aged 35 to 74 years old in 2 Argentinean cities (Bariloche and Marcos Paz) [25]. The questionnaire was based on the Spanish version of the Dietary History Questionnaire I, a self-administered food-frequency questionnaire developed for the Spanish-speaking population in the US [28] and validated for use in Argentina [29]. We assumed that consumption among people 75 years of age and older was equivalent to that reported by those 65 to 74 years old.

We derived a "high" estimate using the 2005 National Nutrition and Health Survey (Encuesta Nacional de Nutrición y Salud [ENNyS]) in combination with sales report data from Euromonitor measured in 2005 and 2015. The ENNyS, a survey conducted in 2005 among Argentine women 10 to 49 years old [30], reported mean daily soda consumption. We applied that figure to women aged 35–44 years in the model. However, recognizing that daily soda consumption includes both regular and diet soda, we estimated the consumption of regular soda by using the fraction of regular soda (87.1%) that makes up sales of total soda (regular

and diet) in the 2005 Euromonitor sales data [10]. Finally, we compared Euromonitor overall sales data in 2015 to 2005 and corrected for the difference in population size between those years. By doing so, we derived a mean daily consumption of regular soda of 269.6 ml for women 35–44 years old in 2015. We then generated estimates of SSB intake for each age/gender group by scaling to the proportional intake observed for each group relative to 35- to 44-year-old women as measured in CESCAS I. A more detailed description of these calculations, as well as a comparison with CESCAS I reported consumption, can be seen in S1 Appendix.

## SSB effect on CVD risk factors

SSB consumption has been shown to have a direct effect on SBP and diabetes, as well as indirect effects on SBP and diabetes that are mediated through changes in BMI [2–4,8]. We modeled both direct and indirect effects of reductions in SSB consumption using inputs shown in Table 1. The model applies all changes in the first year of the simulation and then assumes that they are held constant.

We assumed that decreasing intake of SSB by one 12-ounce serving per day is associated with a reduction in SBP of 0.78 mmHg (95% CI 0.09–1.47) in men and 0.61 mmHg (95% CI 0.27–1.48) in women independent of SBP changes mediated through BMI [31]; similarly, BMI-independent diabetes risk was assumed to decrease by a factor of 1.19 (95% CI 1.09–1.31) [32].

Weight and height information for the Argentina population was obtained from the 2013 National Risk Factor Survey [12], and changes in weight assumed for the interventions were converted into changes in kilograms per square meter of BMI. Each unit kilogram-per-square-meter decrease in BMI was assumed to result in a 1.43 mmHg and 1.24 mmHg decrease in SBP in men and women, respectively [33]. The association between changes in BMI and changes in diabetes incidence was estimated using Framingham Heart Study data [34–38].

We assumed that changes in SSB consumption were associated with changes in weight using the conversion 3,500 kcal = 1 pound; one serving of soda contains 150 calories [39]. Although imbalances between the intake and utilization rates of macronutrients will result in changes in body weight, the relationship between caloric consumption and weight loss/gain, as well as caloric compensation when reducing one source of calories, are not fully understood.

**Table 1. Effect of 12-ounce serving size of SSB on diabetes incidence, BMI, and SBP.**

| Variable | SSB consumption effect |
|---|---|
| SBP; SSB consumption independent effect | Men: 0.78 mmHg (0.09–1.47) |
|  | Women: 0.61 mmHg (0.27–1.48) [31] |
| Independent effect of 1 additional serving of SSB on risk of diabetes, RR (95% CI) | 1.19 (1.09–1.31) [32] |
| SBP; SBP change per 1-unit increase in BMI[a] | Men: 1.43 mmHg (1.23–1.64) |
|  | Women: 1.24 mmHg (1.09–1.39) [33] |
| Association between a 1-unit increase in BMI and incident diabetes, RR (95% CI)[a] | Decreases over age; |
|  | 1.17 (0.97–1.43) for 55–64 years old [34–38] |
| BMI | Calories from SSB translated to weight; |
|  | 3,500 calories = 1 pound [39] |

[a]BMI-mediated changes in diabetes and SBP are expressed for each unit $kg/m^2$ change in BMI.

**Abbreviations**: BMI, body mass index; RR, relative risk; SBP, systolic blood pressure; SSB, sugar-sweetened soda

Consequently, we decided to include 3 scenarios with different degrees of caloric compensation (and therefore different changes in BMI) after a decrease in SSB consumption: 100%, 39% (a reported average compensation rate), and 0% caloric compensation scenarios [40,41]. Under assumptions of 100% calorie compensation (i.e., no change in BMI), the independent effect of SSB consumption on SBP and diabetes drives changes in CVD outcomes.

A more detailed description of our approach to modeling the relationship between changes in SSB consumption and CVD risk factors and outcomes was published in our previous work on SSB in the US and Mexico [42,43].

## Model simulations

We modeled the impact of a reduction in SSB consumption on diabetes incidence, myocardial infarctions, strokes, CVD mortality, and all-cause mortality using the CVD Policy Model-Argentina over a 10-year period from 2015 to 2024.

A recent analysis estimated that the price elasticity of soda consumption in Argentina is very close to 1 (−1.12) [44]. Therefore, we assumed that a 10% or 20% tax could translate into a 10% or 20% decrease in soda consumption (assuming that all, or nearly all, of the cost from the tax is passed on to the consumer) and compared the health impacts of each taxation scenario to the base case of no change in consumption. We also modeled a 40% reduction in SSBs which, though difficult to achieve at a population level, highlights the potential benefits of very high reduction in soda consumption. We applied each taxation scenario to both our "low" and "high" estimates of current SSB consumption in Argentina, and our main analyses assumed 39% calorie compensation [41].

## Probabilistic sensitivity analyses

We used Monte Carlo simulations to generate 95% uncertainty intervals (UIs) around our primary outcome measures for each intervention scenario. The 95% confidence intervals for estimates of the effect of changes in SSB consumption on SBP and on diabetes risk as well as the beta inputs for model risk functions defining the relationship between risk factors and incident diabetes, incident CHD, incident stroke, and non-CVD death are included in Table C of S1 Appendix. There were 1,000 random draws from a standard normal distribution, scaled to the mean and confidence interval, for each varied parameter. The Monte Carlo program, written in Python, generated a new set of input parameters drawn from the distributions for each iteration, ran the given iteration base case and reduced SSB consumption simulations with the new parameters, and stored the outcomes for each iteration. The 95% UIs for each outcome were then calculated using Microsoft Excel 2016.

## Results

Over the 2015–2024 period, a 10% reduction in SSB consumption with a caloric compensation of 39% is projected to reduce diabetes cases by between 13,300 (95% UI 10,800–15,600 [low SSB consumption estimate]) and 27,700 cases (95% UI 22,400–32,400 [high SSB consumption estimate]), i.e., 1.7% and 3.6% fewer cases, respectively, compared to a scenario of no change in SSB consumption. This same scenario leads to an estimated 2,500 (95% UI 2,200–2,800) to 5,100 (95% UI 4,500–5,700) fewer myocardial infarctions, 1,900 (95% UI 1,600–2,200) to 3,900 (95% UI 3,400–4,400) fewer CVD deaths, and 2,700 (95% UI 2,200–3,200) to 5,600 (95% UI 4,600–6,600) fewer deaths from any cause for "low" and "high" estimates of baseline SSB intake, respectively. A 20% reduction in SSB consumption with 39% caloric compensation is projected to result in 26,200 (95% UI 21,200–30,600) to 53,800 (95% UI 43,900–62,700) fewer cases of diabetes, 4,800 (95% UI 4,200–5,300) to 10,000 (95% UI 8,800–11,200) fewer

myocardial infarctions, and 5,200 (95% UI 4,300–6,200) to 11,000 (95% UI 9,100–13,100) fewer deaths (Table 2).

Fig 1 shows the prevented cases of myocardial infarction, stroke, and deaths by gender and age group for the 10% consumption reduction scenario in the 10-year period. The magnitude of prevented cases was highest among men and in the youngest ages modeled, and lowest in the highest ages modeled.

Table 3 presents results from 0% and 100% caloric compensation scenarios, assuming a 10% reduction in SSB consumption. When varying the degree of caloric compensation, even under the most conservative assumption of base SSB consumption (low estimate), the reduction in diabetes ranges from 5,100 (95% UI 3,000–7,100) fewer cases assuming 100% caloric compensation (0.7% relative reduction) to 18,300 (95% UI 15,000–21,200) fewer cases (2.3% relative reduction) if we assume 0% caloric compensation.

The decreased rate of incident diabetes cases under the 6 scenarios of taxation and caloric compensation (10% and 20% consumption reduction, 0%, 39%, and 100% caloric compensation), for low and high estimates of current SSB consumption, can be seen in Fig 2. In each scenario, most of the prevented cases occur in the youngest age groups for both women and men.

In an extreme scenario of achieving a 40% reduction of SSB consumption along with 39% caloric compensation, we estimate 49,800 (95% UI 40,400–58,200) to 101,100 (95% UI 74,200–117,300) fewer cases of diabetes, 9,500 (95% UI 8,400–10,600) to 19,700 (95% UI 16,900–22,000) fewer myocardial infarctions, and from 10,400 (95% UI 8,500–12,300) to 21,600 (95% UI 16,800–25,700) fewer deaths, respectively, for low and high estimates of baseline SSB consumption.

## Discussion

The need to reduce SSB consumption in Argentina is increasingly a matter of policy discussions in response to growing levels of obesity amidst very high levels of SSB consumption.

**Table 2. Projected cumulative cases of diabetes, CVD events, and deaths prevented for the period 2015–2024, assuming 39% caloric compensation, under two scenarios of SSB reduction (10% and 20%) and SSB base consumption estimation (low and high estimates).**

| Preventable cases | Base case total number | 10% reduction, N (%) | | | | 20% reduction, N (%) | | | |
|---|---|---|---|---|---|---|---|---|---|
| | | Low base SSB consumption estimation | | High base SSB consumption estimation | | Low base SSB consumption estimation | | High base SSB consumption estimation | |
| | | N (95% UI) | % change from base case | N (95% UI) | % change from base case | N (95% UI) | % change from base case | N (95% UI) | % change from base case |
| Cases of diabetes | 779,900 | 13,300 | 1.7 | 27,700 | 3.6 | 26,200 | 3.4 | 53,800 | 6.9 |
| | | (10,800–15,600) | | (22,400–32,400) | | (21,200–30,600) | | (43,900–62,700) | |
| Myocardial infarction | 491,700 | 2,500 | 0.5 | 5,100 | 1.0 | 4,800 | 1.0 | 10,000 | 2.0 |
| | | (2,200–2,800) | | (4,500–5,700) | | (4,200–5,300) | | (8,800–11,200) | |
| Stroke | 728,800 | 1,500 | 0.2 | 3,100 | 0.4 | 2,900 | 0.4 | 6,000 | 0.8 |
| | | (1,000–2,000) | | (2,100–4,100) | | (1,900–3,900) | | (4,200–8,200) | |
| CVD deaths | 640,100 | 1,900 | 0.3 | 3,900 | 0.6 | 3,600 | 0.6 | 7,600 | 1.2 |
| | | (1,600–2,200) | | (3,400–4,400) | | (3,200–4,100) | | (6,600–8,800) | |
| Total deaths | 3,309,200 | 2,700 | 0.1 | 5,600 | 0.2 | 5,200 | 0.2 | 11,000 | 0.3 |
| | | (2,200–3,200) | | (4,600–6,600) | | (4,300–6,200) | | (9,100–13,100) | |

CVD deaths: deaths due to CHD + stroke.

**Abbreviations**: CHD, coronary heart disease; CVD, cardiovascular disease; SSB, sugar-sweetened beverage or soda; UI, uncertainty interval

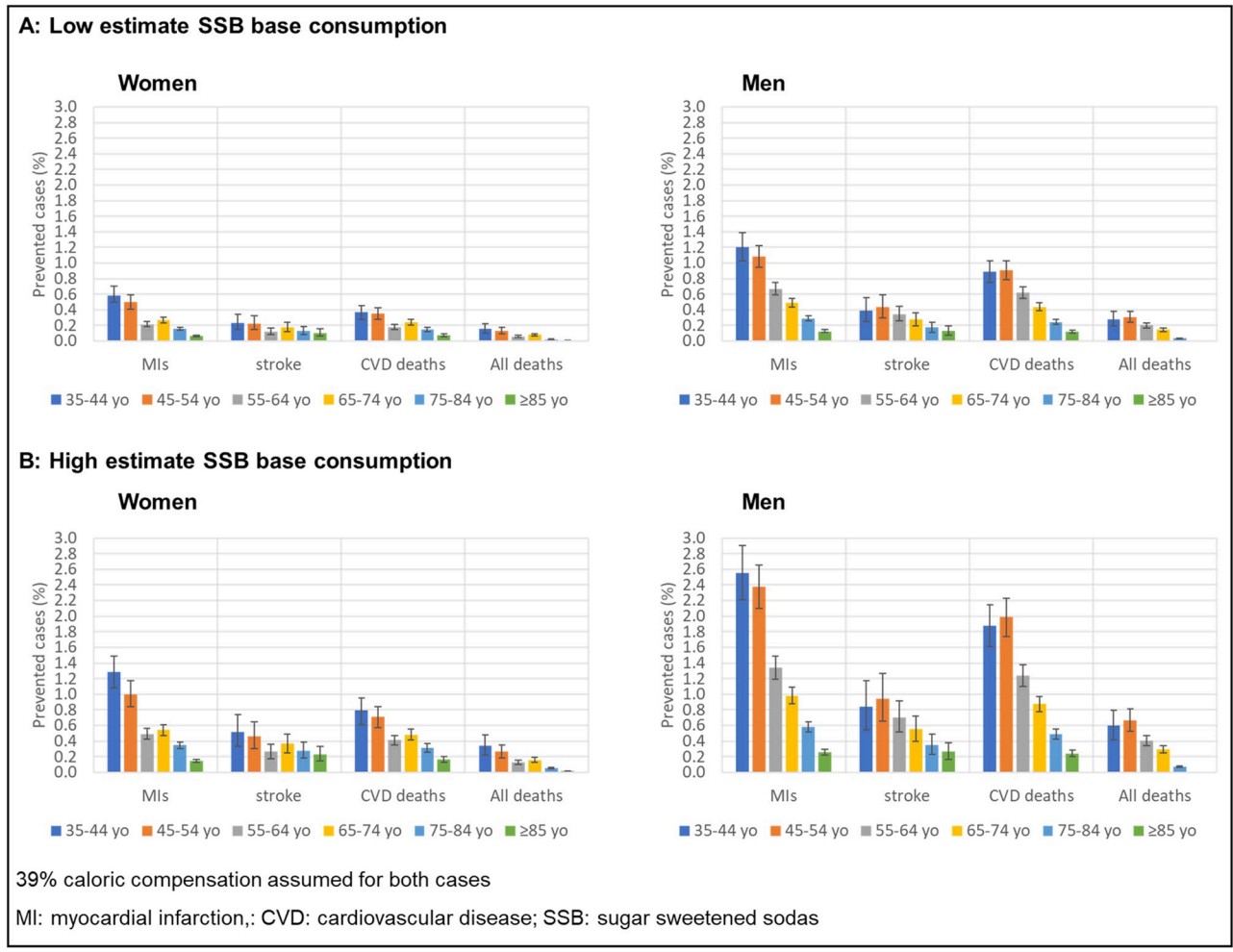

**Fig 1. Projected prevented cases as percent of MI, strokes, CVD deaths, and overall deaths by age group and gender, 2015–2024, assuming 10% reduction in SSB consumption, under two scenarios of baseline SSB consumption.** CVD, cardiovascular disease; MI, myocardial infarction; SSB, sugar-sweetened soda.

Local evidence would best inform that debate. Using a validated computer simulation model populated with Argentine data, we found that a 10% reduction in SSB consumption among those aged 35 years and older could reduce diabetes by a magnitude of 5,100 cases (in an extremely conservative scenario) to 38,200 cases over 10 years, with a most likely impact being between 13,300 to 27,700 cases averted. These last figures would be equivalent to 1 to 2 cases of diabetes avoided for every 1,400 people over 34 years of age. Cardiovascular events and mortality could also be significantly reduced.

It is worth underscoring that, since the average daily consumption of sodas is greater among young people, most prevented cases of diabetes occur in this population. The true health benefits of SSB taxation could be higher than those we present here, as a 10-year estimate of avoided cases of CVD and death is a relatively short time period in which to perceive clinical impact among young and middle-aged adults.

A previous analysis conducted in Mexico using the CVD Policy Model-Mexico found that a 10% decrease in consumption, assuming 39% caloric compensation, could prevent almost 190,000 diabetes cases over a 10-year period, many more than the 13,300 to 27,700 cases resulting

**Table 3. Projected prevented cumulative cases of diabetes, CVD events, and deaths for the period 2015–2024, assuming 10% of SSB consumption reduction, under two scenarios of caloric compensation (0% and 100%) and SSB base consumption estimation (low and high estimates).**

| Preventable cases | Base case total number | Low base SSB consumption estimation | | | | High base SSB consumption estimation | | | |
|---|---|---|---|---|---|---|---|---|---|
| | | 0% caloric compensation | | 100% caloric compensation | | 0% caloric compensation | | 100% caloric compensation | |
| | | N (95% UI) | % change from base case | N (95% UI) | % change from base case | N (95% UI) | % change from base case | N (95% UI) | % change from base case |
| Cases of diabetes | 779,900 | 18,300 (15,000–21,200) | 2.3 | 5,100 (3,000–7,100) | 0.7 | 38,200 (31,300–44,000) | 4.9 | 10,500 (6,200–15,000) | 1.3 |
| Myocardial infarction | 491,700 | 3,800 (3,400–4,100) | 0.8 | 250 (200–350) | 0.1 | 7,800 (7,000–8,700) | 1.6 | 500 (200–900) | 0.1 |
| Stroke | 728,800 | 2,100 (1,500–2,800) | 0.3 | 400 (300–550) | 0.1 | 4,400 (3,100–5,800) | 0.6 | 900 (200–1,600) | 0.1 |
| CVD deaths | 640,100 | 2,800 (2,500–3,100) | 0.4 | 250 (200–300) | 0.05 | 5,900 (5,200–6,500) | 0.9 | 500 (200–900) | 0.1 |
| Total deaths | 3,309,200 | 3,900 (3,300–4,500) | 0.1 | 600 (450–750) | 0.02 | 8,200 (6,900–9,500) | 0.2 | 1,300 (500–2,000) | 0.04 |

CVD deaths: deaths due to CHD + stroke.

**Abbreviations**: CHD, coronary heart disease; CVD, cardiovascular disease; SSB, sugar-sweetened beverage or soda; UI, uncertainty interval

from the same scenario in Argentina. This large difference is likely due to the size of Mexico's population (almost 3 times larger than Argentina's) [45,46], a higher average daily SSB consumption in the Mexico analysis (that additionally included all sugary drinks), and a current higher rate of obesity in Mexico. In 2013, Mexico implemented an excise tax on soft drinks; estimates

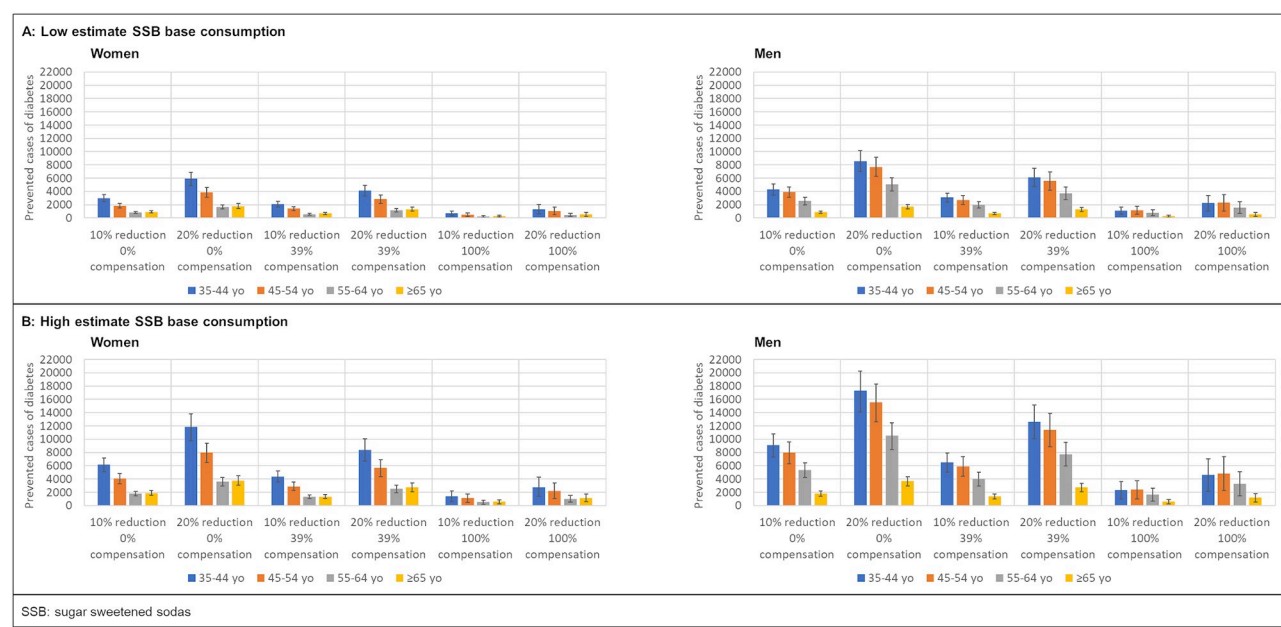

**Fig 2. Projected prevented cases of diabetes by age group and gender for the period 2015–2024 under 6 different scenarios of SSB reduction in consumption and caloric compensation and 2 estimates of baseline SSB consumption.** SSB, sugar-sweetened soda.

after year 2 indicate that taxed beverages per capita sales were reduced by 7.3%, while per capita sales of plain water increased 5.2% [18]. Mexico's experience informs what Argentina could foresee as expected benefits of a tax policy aimed at reducing SSB consumption.

Our study limitations are mainly related to the limited availability of SSB consumption data in Argentina. We have undertaken several strategies to obtain valid estimates given these limitations, and we present a range of assumptions. The SSB daily serving size as reported by the CESCAS I study (which provides our "low" intake estimate) is likely to be underestimating total consumption. This would bias our results toward a more conservative estimate of health benefits. We have also modeled the health benefits of a consumption reduction using an alternative, less conservative, estimate of current SSB intake in the population. Additionally, our main scenario assumes a 39% caloric compensation; this estimate comes from US data [41] and may be higher or lower in Argentina or could vary in a systematic way over time. Also, the effect size and all model parameters are held constant, varying only by age as the population ages, which is both a strength and a limitation of modeling. To account for these possibilities, we have also modeled the 2 extremes cases of 0% and 100% caloric compensation. Finally, we were only able to model the possible benefits for the population older than 34 years—the age range analyzed by the CVD Policy Model—and for a 10-year period. As most sodas are consumed by younger people, the health benefits of consumption reduction could be higher among younger generations over time.

Despite uncertainty about the distribution of SSB consumption among Argentina's population, the government will need to make public health policy decisions about whether and how to limit SSB consumption. Our study finds that, even under conservative assumptions, a relatively small reduction in SSB consumption could lead to a significant decrease in diabetes incidence, CVD events, and mortality. These results support policies to increase the price of these products using taxation as a potential tool to reduce SSB consumption, which would have the additional advantage of providing a new source of public funds to support healthy lifestyles.

Argentina has previously used computer simulation research results to foster national policy development. For example, other modeling studies on tobacco control [47] and salt consumption [48] (using the CVD Policy Model-Argentina) highlighted the impact of potential policies that were subsequently implemented. The results of this study should contribute to the development and implementation of evidence-based policies aimed at decreasing SSB consumption in Argentina.

## Supporting information

**S1 Appendix. The Cardiovascular Disease Policy Model.** Update and calibration of CVDPM-Argentina. SSB consumption estimations. Table A: Local data sources for CVD Policy Model-Argentina update and calibration. Table B: Comparison of overall outcomes between model predictions (CVDPM-Arg) and actual statistics in Argentina, 2010. Table C: Summary of variables used for the forecasting of the effect of SSB taxation in Argentina on diabetes, CVDs, and mortality outcomes. Table D: Daily per capita consumption of SSBs in Argentina, by age group and gender, by self-report from 2 cities, 2010–2011. Table E: Comparison of 2 different methodologies for estimating SSB per capita daily consumption in Argentina. CVD, cardiovascular disease; CVDPM-Arg, Cardiovascular Disease Policy Model-Argentina; SSB, sugar-sweetened beverage or soda.
(DOCX)

**S1 Fig. Cardiovascular disease (CVD) Policy Model structure.**
(TIF)

**S2 Fig. Directed acyclic graph describing the relationship between changes in SSB consumption and risk factors and outcomes in the CVD Policy Model-Argentina.** CVD, cardiovascular disease; SSB, sugar-sweetened beverage.
(TIF)

**S1 Protocol. Prospective protocol from funding proposal.**
(PDF)

**S1 STROBE Checklist. Checklist of items that should be included in reports of observational studies.** STROBE, Strengthening the Reporting of Observational Studies in Epidemiology.
(DOCX)

## Author Contributions

**Conceptualization:** M. Victoria Salgado, Joanne Penko, Alicia Fernandez, Jonatan Konfino, Kirsten Bibbins-Domingo, Raul Mejia.

**Formal analysis:** M. Victoria Salgado, Joanne Penko, Jonatan Konfino, Pamela G. Coxson.

**Funding acquisition:** M. Victoria Salgado, Alicia Fernandez, Jonatan Konfino, Raul Mejia.

**Investigation:** M. Victoria Salgado, Joanne Penko, Pamela G. Coxson, Raul Mejia.

**Methodology:** M. Victoria Salgado, Joanne Penko, Alicia Fernandez, Pamela G. Coxson, Kirsten Bibbins-Domingo, Raul Mejia.

**Project administration:** Alicia Fernandez, Raul Mejia.

**Resources:** Raul Mejia.

**Supervision:** M. Victoria Salgado, Alicia Fernandez, Raul Mejia.

**Writing – original draft:** M. Victoria Salgado, Joanne Penko, Alicia Fernandez, Jonatan Konfino, Pamela G. Coxson, Kirsten Bibbins-Domingo, Raul Mejia.

**Writing – review & editing:** M. Victoria Salgado, Joanne Penko, Alicia Fernandez, Jonatan Konfino, Pamela G. Coxson, Kirsten Bibbins-Domingo, Raul Mejia.

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
