## [Editor Report · Decision Letter 0]

11 Feb 2020

Dear Dr Salgado, 

Thank you for submitting your manuscript entitled "Projected impact of a reduction in sugar-sweetened beverages consumption on diabetes and cardiovascular disease in Argentina: a modeling study" for consideration by PLOS Medicine.

Your manuscript has now been evaluated by the PLOS Medicine editorial staff [as well as by an academic editor with relevant expertise] and I am writing to let you know that we would like to send your submission out for external peer review.

Kind regards,

Adya Misra, PhD,

Senior Editor

PLOS Medicine

---

## [Decision Letter · Decision Letter 1]

15 May 2020

Dear Dr. Salgado,

Thank you very much for submitting your manuscript "Projected impact of a reduction in sugar-sweetened beverages consumption on diabetes and cardiovascular disease in Argentina: a modeling study" (PMEDICINE-D-20-00333R1) for consideration at PLOS Medicine. 

[LINK]

In light of these reviews, I am afraid that we will not be able to accept the manuscript for publication in the journal in its current form, but we would like to consider a revised version that addresses the reviewers' and editors' comments. Obviously we cannot make any decision about publication until we have seen the revised manuscript and your response, and we plan to seek re-review by one or more of the reviewers. 

We expect to receive your revised manuscript by Jun 05 2020 11:59PM. Please email us (plosmedicine@plos.org) if you have any questions or concerns.

We look forward to receiving your revised manuscript. 

Sincerely,

Emma Veitch, PhD

PLOS Medicine

On behalf of:

Adya Misra, PhD

Senior Editor 

PLOS Medicine

plosmedicine.org

*Just to note that this paper is being considered for the journal's special issue on Obesity - please do liaise with Adya regarding any points regarding inclusion of this article in the special issue, and bear in mind that being able to revise in line with our deadlines will help ensure inclusion, if the article passes peer review.

*As noted by one reviewer, addressing more fully the range of parameter uncertainty in the model would be helpful, although we recognise this would represent a substantial amount of reworking of the paper and presentation of the findings. However we look forward to the authors' justification on this point, and if the authors choose not to address this in the way suggested by the reviewer, we feel a strong justification would be needed. 

*Please structure your abstract using the PLOS Medicine headings (Background, Methods and Findings, Conclusions- "Methods and Findings" should be a single subsection). 

*At this stage, we ask that you include a short, non-technical Author Summary of your research to make findings accessible to a wide audience that includes both scientists and non-scientists. The Author Summary should immediately follow the Abstract in your revised manuscript. This text is subject to editorial change and should be distinct from the scientific abstract. Please see our author guidelines for more information: https://journals.plos.org/plosmedicine/s/revising-your-manuscript#loc-author-summary

*In the last sentence of the Abstract Methods and Findings section, please describe the main limitation(s) of the study's methodology.

Comments from the reviewers:

Reviewer #1: This is a statistical review of manuscript PMEDICINE-D-20-00333_R1. The manuscript reads well. I do not have any specific comment on this manuscript in its current state. 

However, I have an important suggestion that might improve the manuscript. In its current state, the modelling exercise does not take into account the uncertainty in the parameters of Table 1. For example, the effect on systolic blood pressure is 0.78 mmHg (95% CI 0.09 to 1.47), but the authors use 0.78 mmHg in the model, as shown on Figure S2. A more realistic modelling exercise would use Monte-Carlo simulations to assess the reduction in number of events by taking into account the uncertainty on all parameters of Table 1 (diabetes, fatal and non-fatal cvd events, mortality). Using Monte-Carlo simulations, the authors would be able to provide a range for their estimates (and present this visually using histograms) for the various scenarios (low vs high consumption, caloric compensation of 0% , 39% and 100%). 

I recognise that my suggestion will generate more work for the authors. Nonetheless I think that providing uncertainty intervals for the estimates would noticeably improve the paper. For example, the sentence in the abstract that currently reads "a 10% reduction in SSB with a caloric compensation of 39% is projected to reduce incident diabetes cases by 13300 to 27700 cases compared to a scenario of no change in SSB consumption" would include uncertainty intervals around 13300 and 27700. Given the uncertainty in the effects of the intervention (as recognised in Table 1) it would be more appropriate to propagate the uncertainty in the modelling exercise. 

Reviewer #2: The CVD policy model-Argentina assessed the impact of an SSB taxation in Argentina, it models a lower and a high-consumption scenario with different levels of compensation from other beverages. The paper shows that even if with 10% reduction in soda consumption from the Argentinian population an important number of diabetes and CVD cases can be avoided. The clearly stated the assumptions of the model and limitations of the available data

This is a clear and well written paper. Obesity in Argentina is a great public health problem. I believe that the results from this paper will provide good evidence to support the implementation of a soda taxation. Results from this simulation model has been previously used for policy advocacy, in Mexico, US for SSB tax and Argentina for Salt reduction. Using Simulation Modeling for assessment of regulatory actions is becoming a very popular tool for the health research community not only for policy but also as a method to assess the epidemiological data needed for a better evaluation. 

This work is likely to mater to the public health community who will translate the results to disseminate for policy makers and other Ministry of Health stakeholders. Given the implementation of SSB tax all over the world this study is of likely interest to all readers. 

Minor comment. 

Abstract: The line before conclusions, please clarify at which age gender you refer when saying." Cardiovascular events occurred in the youngest age group (35-44 y)" It is confusing as you mentioned afterwards that the reduction would also occur in men all age. 

Reviewer #3: 

Projected impact of a reduction in sugar-sweetened beverages consumption on diabetes and cardiovascular disease in Argentina: a modeling study"

PLos Medicine

This is a very good simulation exercise yielding much needed estimations for Argentina about the potential impact of raising taxes on SSBs on important outcomes as CVD morbidity and mortality.

They used a very well-known Markov model developed by L. Goldman and M. Weinstein, based on those equations and risk estimations derived from the Framingham cohort study.

I would like to praise the authors for making an importan effort in identifying appropriate locally relevant parameters, about consumption and sales, caloric compensation, and relative risks for selected conditions. It is also a strength to consider low- and high- consumption, and three taxing scenarios.

They focused on CVD outcomes, due to the nature of the model they selected, which is only part of the global SSB public health picture. It is important to note that for SSBs as a health risk factor, the population of children, adolescents and young adults is the most affected and this is not adequately captured by this model. Also, other relevant public health outcomes such as dental caries, bullying, depression, cancer of many sites, and many medical conditions such as non-ischemic heart disease, osteoarthritis, kidney disease, dementia and others, could be incorporated in future SSB-specific modeling. The discussion should have highlighted this wider panorama. From the content point of view, what seems to be also missing is a clear and wider explanation of ‎strengths/weaknesses, particularly in context of other concurrent measures to be taken in NCD prevention.

Overall, the manuscript has potential to soundly modify health policy, especially if coupled with other more SSB-specific simulations. I strongly recommend this paper to be published.

Minor comments: 

Ref11 is a short newspaper article, not suitable as a source for a manuscript in my opinion. Better to original sources. Euromonitor is referenced elsewhere in the manuscript.

Ref 41 for the estimate of calorie compensation is rather old.

doi: 10.1038/ijo.2015.177 This meta-analysis from Rogers et al could be helpful. However the authors correctly considered two extreme cases to account for this.

The authors assume that all cities in Argentina consume soda in the same way of Bariloche and Marcos Paz, hopefully during 2020 the second ENNyS survey will be published and more representative data is available. In the meantime, I think what they did is OK and is properly discussed.

Table 1. Some relative risks (RRs) might have been more updated or better substantiated by evidence. For example, the Global Burden of Disease 2017 study, available from http://ghdx.healthdata.org/gbd-results-tool would be an alternative and excellent source for deriving updated RRs

S2 table: what are exactly MI Deaths and arrests? what ICD10 codes were considered? There is no mention of corrections for garbage codes, or under-reporting of CV deaths, although it seems to have been done, and stated in the Medicina companion paper (Ref 27), I think it is worth to mention this in the main manuscript and in the Supplementary file.

BMI effects on CVD events independent of DBT and lipids are not considered, possibly making the model estimates even more conservative. See for example DOI: 10.1001/archinte.167.16.1720 This could be discussed.

Also, the model applied changes in the first year and then assumes to hold constant which could be questionable. 

Other technical comments: 

The term 'diabetes' should be replaced by 'Type 2 Diabetes Mellitus' which is more accurate.

Table 2: it wold be clearer to say CVD deaths instead of CVD mortality (which is a rate)

Fig 2 almost unreadable, need to fix the resolution of the image.

[LINK]

---

## [Editor Report · Decision Letter 2]

8 Jun 2020

Dear Dr. Salgado,

Thank you very much for re-submitting your manuscript "Projected impact of a reduction in sugar-sweetened beverages consumption on diabetes and cardiovascular disease in Argentina: a modeling study" (PMEDICINE-D-20-00333R2) for review by PLOS Medicine.

I have discussed the paper with my colleagues and the academic editor and it was also seen again by reviewers. I am pleased to say that provided the remaining editorial and production issues are dealt with we are planning to accept the paper for publication in the journal.

[LINK]

We look forward to receiving the revised manuscript by Jun 15 2020 11:59PM. 

Sincerely,

Adya Misra, PhD

Senior Editor 

PLOS Medicine

plosmedicine.org

Requests from Editors:

Abstract needs additional background to provide context for the study

Some tempering of language in the abstract and throughout the submission- “would” and “were” should be changed to “could”, “occurred” should be changed to “observed” to reflect the study design and observational nature of your study

Please add a space between text and reference square brackets followed by a full stop throughout.

Throughout the manuscript please avoid the use of stigmatising labels such as “overweight” or “obese” and instead use “unhealthy weight” and “with obesity”

Line 111 please revise to “non-communicable” disease

Line 154, 158 “gender” should be revised to “sex”

Prospective analysis plan

Table 2,3- please add “projected” when you mention prevented as this is a modelling study. The same goes for figures and captions

Please ensure that the study is reported according to the STROBE guideline, and include the completed checklist as Supporting Information. When completing the checklist, please use section and paragraph numbers, rather than page numbers. Please add the following statement, or similar, to the Methods: "This study is reported as per the Strengthening the Reporting of Observational Studies in Epidemiology (STROBE) guideline (S1 Checklist)."

Did your study have a prospective protocol or analysis plan? Please state this (either way) early in the Methods section.

- it looks like the model has been published previously, but I'm not sure "contact the authors" in the data statement complies with our usual policy - that might be best removed

- "per capita" at line 26

- around line 40, it would be helpful to quote the projected percentage reduction in incident T2D cases, as at line 252

- At lines 56 and 85, suggest replacing "significant" with "substantial" (or similar)

- At line 76, "projected to decrease" is not quite grammatical (e.g., "projected to lead to a decrease")

- "Discussion" at line 296

- At line 302 where the authors summarize the findings, they start by noting a conservative projected reduction of 5100 T2D cases, which I don't think is quoted in the abstract or author summary (it does appear in the results). I'd suggest they restructure this to avoid confusing readers

Comments from Reviewers:

[LINK]

---

## [Editor Report · Decision Letter 3]

22 Jun 2020

Dear Dr. Salgado, 

On behalf of my colleagues and the academic editor, Dr. Sanjay Basu, I am delighted to inform you that your manuscript entitled "Projected impact of a reduction in sugar-sweetened beverages consumption on diabetes and cardiovascular disease in Argentina: a modeling study" (PMEDICINE-D-20-00333R3) has been accepted for publication in PLOS Medicine. 

PRODUCTION PROCESS

PRESS

PROFILE INFORMATION

Thank you again for submitting the manuscript to PLOS Medicine. We look forward to publishing it. 

Best wishes, 

Adya Misra, PhD

Senior Editor 

PLOS Medicine

plosmedicine.org